# Abundance and morphology of charcoal in sediments provide no evidence of massive slash-and-burn agriculture during the Neolithic Kuahuqiao culture, China

**YuanFeng Hu**[1], **Bin Zhou**[1]\*, **YueHan Lu**[2], **JianPing Zhang**[3], **SiYu Min**[1], **MingZhe Dai**[1], **SiYu Xu**[1], **Qing Yang**[4], **HongBo Zheng**[4]

**1** Key Laboratory of Surficial Geochemistry, Ministry of Education, School of Earth Sciences and Engineering, Nanjing University, Nanjing, China, **2** Molecular Eco-Geochemistry Laboratory, Department of Geological Sciences, University of Alabama, Tuscaloosa, Alabama, United States of America, **3** Key Laboratory of Cenozoic Geology and Environment, Chinese Academy of Science, Institute of Geology and Geophysics, Chinese Academy of Sciences, Beijing, China, **4** Yunnan Key Laboratory of Earth System Science, School of Resource, Environment and Earth Science, Yunnan University, Chenggong District, Kunming, China

\* zhoubinok@163.com

**Data Availability Statement:** All relevant data are within the manuscript and its Supporting Information files.

## Abstract

It remains debatable whether slash-and-burn practices were adopted in rice cultivation by the Neolithic Kuahuqiao culture in the Ningshao Plain, one of the birthplaces of rice farming. Here, we established charcoal-based indices to reconstruct the history of fire and vegetation in the Ningshao Plain since the last glacial period. We collected representative modern vegetation and conducted combustion and fragmentation experiments to simulate fire and depositional processes, respectively. Charcoals from modern vegetation show clear morphological differences between herbaceous and woody plants. In particular, the length to width ratios (L/W) of herbaceous charcoals were systematically higher than those of woody charcoals, and the associated end-member values were 4.50 and 1.94, respectively. These values were then applied to sediment cores (KHQ-14/15) collected in proximity to the Kuahuqiao archaeological site. Results show that the amount of combusted herbaceous plants increased sharply after the Holocene, and the most remarkable rise occurred around 8550 yr B.P. This observation may reflect local environment (sedimentary and/or climatic) changes or small-scale early human activities. During the Kuahuqiao cultural period (8250−7450 yr B.P.), the relative abundance of woody charcoals increased, but the overall fire intensity decreased. This finding suggests that the Kuahuqiao farming was restricted to a small geographic area and large-scale slash-and-burn farming activities were not adopted.

## Introduction

The Ningshao Plain located in the eastern part of China is home for advanced prehistoric civilization and one of important birthplaces of rice farming. Kuahuqiao in the Ningshao Plain

**Funding:** The National Natural Science Foundation of China (41977378), Major Research Plan of the National Natural Science Foundation of China (Grand No. 41991323), Research on the Global Change of National Major Scientific Research Project of China (2015CB953804), and Jiangsu Provincial Basic Research Program Natural Science Foundation General Project of China (BK20171340) had play an important role in our study design, data collection and analysis, decision to publish.

**Competing interests:** The authors have declared that no competing interests exist.

has been considered as a key location to study the early Neolithic rice culture [1–4]. It is commonly accepted that fire was widely used in agricultural management in this early culture [5], yet it remains unclear whether the slash-and-burn agricultural model was adopted. For example, charcoal and pollen data from the Kuahuqiao site show that Neolithic settlers in the Ningshao Plain used fire to clear alder-dominated wetlands for early rice cultivation, and the first rice cultivation occurred around 7700 yr B.P. [5–7]. However, due to the low charcoal contents from nearby wetland sedimentary profiles, Shu et al. [7] suggested that the charcoal in the Kuahuqiao site may be produced from daily life activities (e.g., cooking), rather than large-scale fire to clear mountainous forest and prepare ground for early rice cultivation.

The inconclusive results of the reconstructed fire history of the Ningshao Plain [5, 7–13] can be largely attributed to the decoupling between the vegetation type indicated by pollen and vegetation combustion indicated by charcoals. It is thus critical to establish proxies that can unambiguously link the type and combustion of vegetation, i.e., identifying the type of combusted vegetation. In the present study, we established such indices based on charcoal morphology, and we applied these indices to determine whether and when massive burning of native plants occurred during the period of the Kuahuqiao culture. Our primary object was to test the hypothesis that the slash-and-burn model was used for rice cultivation during the Kuahuqiao culture.

Charcoals are the most commonly used proxy to reconstruct fire history. They are a black mixture of graphite and organic carbon compounds resulting from incomplete combustion (or pyrolysis) of plant tissues [14–19]. Charcoals are widely observed in sediments, resistant to post-burial changes [20–24], and thus frequently used to reconstruct paleo-fire history and associated climate and vegetation changes [13, 18, 25, 26]. The abundance of charcoals in sediments can indicate the intensity and frequency of fires [14, 17], whereas the size of charcoals reflects, to a certain extent, the distance of fire sources. Large charcoal particles (usually >125 μm) do not travel far and thus indicate local fire activities, generally within a range of 7 km [14, 17], while micro-sized charcoals (<125 μm) can indicate fire events at a regional scale [14, 17]. The form and structure of charcoals may indicate the type of plants (woody versus herbaceous) that were combusted [17, 24, 27–31]. For example, the length to width ratios of charcoals (L/W) in loess were used to identify the type of vegetation over the last interglacial period, yielding results that compare favorably with those from pollen [32, 33].

The deposition and preservation of charcoals in sediments are subjected to transportation and fragmentation that can vary as a function of vegetation type and geographic variability. Therefore, we established charcoal-based paleofire indices from comparable modern vegetation and environment. We performed combustion and fragmentation experiments on representative types of modern vegetation collected in the Ningshao Plain. We further applied these indices to wetland sediment cores to establish the evolution history of ancient fire and vegetation in the Ningshao Plain since the last glacial period, including the Kuahuqiao cultural period. Results from this study offer new insight into the role of fire during the development of Neolithic agricultural activities in the context of natural environmental change.

## Materials and methods

### Study sites and sample collection

The Ningshao Plain is an east-west extending, long, and narrow coastal plain on the south bank of the Hangzhou Bay (Fig 1). The plain is bounded by the East China Sea to the east, the East Siming hills to the south, the Southwest Tianmushan hills and mountains to the west, and the Hangzhou Bay and Hangjiahu plain to the north. The study area is situated near the northern boundary of the central subtropical climate zone, where the climate is highly influenced by

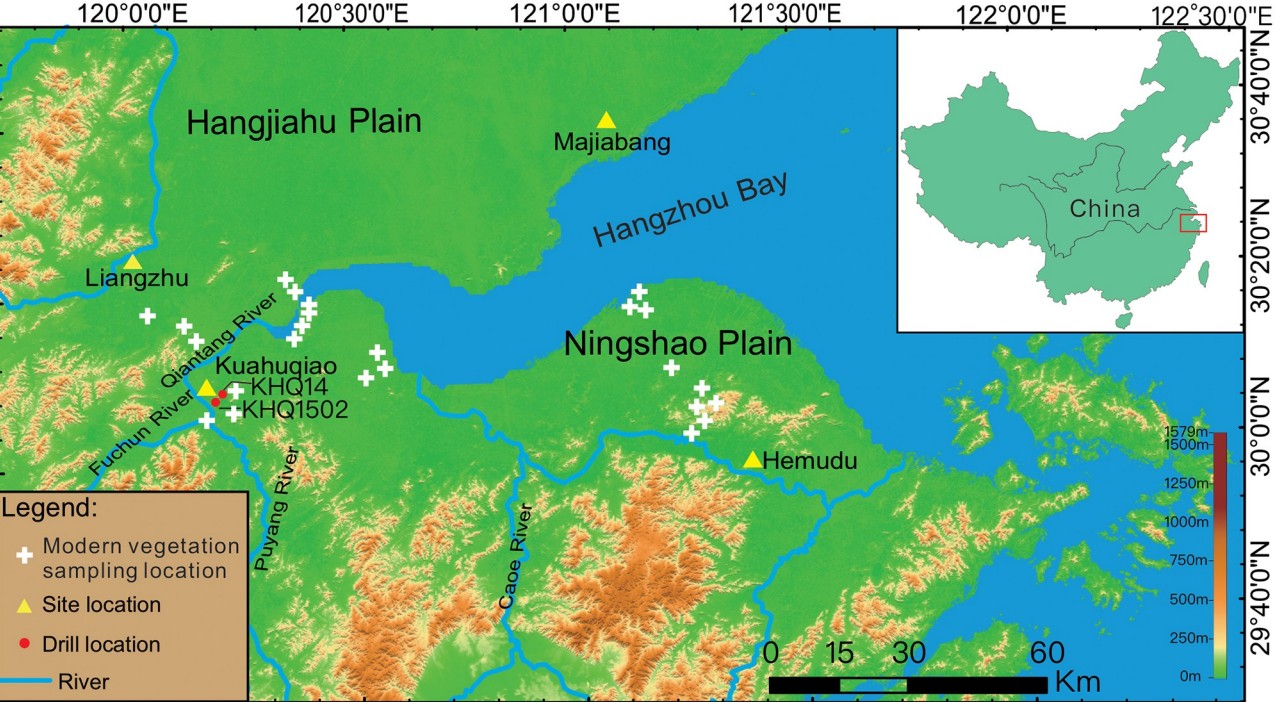

**Fig 1. The study site map.** The Kuahuqiao area is located in the intersection of the Qiantang River, Fuchun River, and Puyang River; the Kuahuqiao archeological site (30° 08'42" N, 120° 13'02″E) is situated at the northeastern part of the Kuahuqiao area. White crosses denote the vegetation sampling locations for combustion experiments; yellow triangles denote main archeological sites in this area, and red circles represent the locations of two sediment cores (KHQ14 and KHQ 1502).

East Asian Monsoon, with high temperature and precipitation in summer and low temperature and precipitation in winter. The mean annual temperature is ca. 16°C, and the mean annual precipitation is ca. 1460 mm. The local modern vegetation is dominated by central subtropical evergreen deciduous broad-leaved mixed forest composed mainly of *Quercus*, *Pinus*, *Alnus*, *Cyclobalanopsis*, Gramineae, *Artemisia Linn*.

To develop charcoal indices, we collected modern plant samples of representative vegetation types near the Kuahuqiao and the Hemudu archeological sites in the Ningshao Plain (Fig 1). Both sites are known for developing the earliest known rice agriculture [2, 5]. We collected grass species that included *Miscanthus*, *Echinochloa*, *Setaria*, Leaves of *Phyllostachys*, *Arundo donax*, *Phragmites*, *Panicum miliaceum*, *Oryza*, as well as tree species that included *Cinnamomum*, *Quercus*, Taxodiaceae, *Ficus microcarpa*, *Liquidambar*, *Artemisia* and a mixture of the trunk of arbor including *Cinnamomum and Quercus* (simplified as Wood), Strip of *Phyllostachys*. Weeds (unidentified species) included two groups collected from tidal flats and paddy fields, respectively.

The Kuahuqiao archeological site (30°08'42" N, 120°13'02" E) is located near the conjunction of the Qiantang River, Fuchun River, and Puyang River (Fig 1). Two sediment cores (KHQ-14 and KHQ-1502) were collected from a wetland (30°07'29" N, 120°11'33" E) that is around 1.5 km southwest of the Kuahuqiao site. The KHQ-14 core was drilled in March 2015, and the KHQ-1502 was drilled in August 2015. Coring was carefully carried out by a professional drilling company to minimize sediment disturbance. The studied area is a relatively closed basin with a stable sedimentary environment, and the two cores are about 10m apart. The core KHQ-14 is 34.00 m in length, and the KHQ-1502 is 20.10 m. Because the top layer (~2.5 m) of KHQ-14 was disturbed by modern human activities, KHQ-1502 was used for

paleoenvironmental interpretation of the upper 18 meters of sedimentary records and the KHQ-14 core was used for the interpretation between 18.00 and 34.00 m. This approach assumed similar stratigraphic and sedimentary changes between the two cores, and our field observations confirmed this assumption.

The sediment cores are mainly composed of clay and silt with clear parallel beddings. We divided the cores into four sedimentary units. The first, bottom unit (from 34.00 to 26.30 m) is a hard clay layer likely representing paleosol deposits during the last glacial period. The second unit has a depth of 26.30–18.40 m, showing rhythmic, interactive deposition (tidal flat facies) of clay and silt. The third unit (18.40–9.50 m) is composed of clay with typical estuarine marine facies, and the fourth, uppermost unit (7.60 m to the surface) is mainly yellow brown sandy silt indicating strong dynamic flooding. For age determination of the sediment cores, seven plant residues of KHQ-1502 and two plant residues of KHQ-14 were submitted to the Beta laboratory (U.S.A.) for Accelerated Mass Spectrometer [14]C dating. Age correction was performed using the INTCAL13 curve correction in OxCal online software (OxCal v4.3.2). The associated procedure and results are described in detail in Yang et al. [34] (under review).

## Combustion experiment

All samples were air-dried at the room temperature (~20˚C), and branches, leaves, and stems were selected for open-fire combustion experiments. Charcoal analysis followed the method described in detail by Zhang & Lv [32]. Briefly, after open-fire combustion, charcoals less than 500 μm were manually separated out under an Olympus BX51TF microscope. These smaller charcoal grains were thoroughly mixed with a glycerin and water mixture (volume: volume = 1:1) and separated by centrifugation. The upper layer was decanted, and the charcoal residues were mounted onto glass slides for observation and measurement under an Olympus BX51TF microscope at the magnification of 200 times. Length (L), L/W, and morphological characteristics (e.g., pore presence, edge shape) of charcoal grains produced by various plants were examined and recorded.

## Charcoal fragmentation experiment

In order to assess the changes in the morphological characteristics of charcoals under various depositional conditions, we used a vibrator (Shanghai Luxi WH-2 micro vortex mixer) and centrifuge (Shanghai Anting DL-5-B low-speed centrifuge) to simulate charcoal fragmentation during sedimentary processes. We first determined the influence of duration of shaking and centrifugation on charcoal morphology. Charcoals from two woody plants, *Quercus* and Taxodiaceae, and two herbaceous plants, Weeds in paddy fields and *Phragmites*, were selected for this experiment. These charcoals were added to blank, organic matter-free soils that were made via high-temperature combustion (650 ˚C for 4 hours). Three vibration and centrifugal conditions were applied to the four charcoal-soil mixtures (Table 1): (i) shaken for 1.5 minutes and then centrifuged for 30 minutes; (ii) shaken for 3 minutes and centrifuged for 60 minutes; and (iii) shaken for 5 minutes and centrifuged for 90 minutes. The vibrator frequency was set at 50 Hz and the centrifuge speed was 2500 rpm.

Based on the results from the above experiments, we selected appropriate shaking and centrifugation conditions and applied these conditions to charcoals obtained from the combustion of typical plants in the study area. Selected plants included *Miscanthus*, Weeds from tidal flats, Weeds from paddy fields, *Echinochloa*, *Setaria*, Leaves of *Phyllostachys*, *Arundo donax*, *Phragmites*, Strip of *Phyllostachys*, *Panicum miliaceum*, *Oryza*, *Cinnamomum*, *Quercus*, Taxodiaceae, *Ficus microcarpa*, *Liquidambar*, Wood and *Artemisia*. After fragmentation, samples were extracted using the method described below (see *charcoal extraction from sediment*

**Table 1. Changes in charcoal morphological indices due to vibration and centrifugation durations.**

| Specie name | Parameter | After combustion experiment | After fragmentation experiments vibration (min)/ centrifugation(min)* | | |
| --- | --- | --- | --- | --- | --- |
| | | | **1.5/30** | **3/60** | **5/90** |
| *Quercus* | L/μm* | 128.0±8.2 | 103.2±8.1 | 94.3±7.3 | 88±7.4 |
| | L/W* | 3.2±0.3 | 2.2±0.3 | 1.7±0.4 | 1.6±0.2 |
| Taxodiaceae | L/μm | 125.1±10.5 | 114.5±6.5 | 90.7±9.5 | 89.5±8.3 |
| | L/W | 2.8±0.2 | 1.8±0.2 | 1.7±0.3 | 1.5±0.3 |
| Trees | L/μm | 126.5±9.35 | 108.9±5.6 | 92.5±7.2 | 88.8±6.3 |
| | L/W | 3.00±0.3 | 2±0.2 | 1.7±0.3 | 1.5±0.2 |
| Weeds from paddy fields | L/μm | 106.1±12.1 | 143.6±9.6 | 116.2±7.6 | 109.5±5.9 |
| | L/W | 9.6±0.4 | 4.4±0.4 | 3.7±0.3 | 3.2±0.2 |
| *Phragmites* | L/μm | 102.1±10.9 | 128.5±10.8 | 111.5±8.4 | 107±8.2 |
| | L/W | 7.6±0.5 | 4.6±0.3 | 4.4±0.3 | 3.7±0.2 |
| Herbs | L/μm | 104.1±11.5 | 136.1±8.3 | 113.8±7.2 | 108.3±5.4 |
| | L/W | 8.6±0.45 | 4.5±0.4 | 4.1±0.3 | 3.5±0.2 |

*all values are presented as mean ± standard deviation; each data point was calculated from 29–50 charcoal grains.

*cores*), and the extracted debris were mounted onto movable slides. The morphology of charcoal and the corresponding parameters were analyzed under a microscope. Morphological characteristics of charcoals of different vegetation types and the L/W end-member values for woody and herbaceous vegetation were determined.

## Charcoal extraction from sediment cores

A total of 16 samples were selected from the four lithologically different units of the sediment cores. Three grams of sediment samples were mixed with 5 ml of 5% sodium hydroxide solution under 70 ˚C for 15 minutes to remove organic acids from the sample. Five milliliters of 10% HCl were added to remove carbonates, followed by three washes with deionized water. Solid residues were sieved through 125 μm, and charcoal particles larger than 125 μm were dried and mounted on glass slides for microscopic examination. The amount of charcoals was counted under a microscope, and L and L/W were recorded.

## Results

### Morphology of charcoals from open-fire combustion of modern plants

Results from the combustion experiments show that charcoal morphology differed across plant types (Fig 2). Charcoals produced from herbaceous plants mostly had an elongated and needle shape with clearly-defined edges, and fractures and pores were present (charts 1–10 in Fig 2). Their L was 125.1±11 μm (mean ± standard deviation) and L/W was 7.55±4.7 (n = 480). Shrub charcoals were mostly rectangular with poorly-defined edges and corners, and they showed coarse fibers on fractured surfaces (chart 13 in Fig 2). Their L and L/W were 127.4±55 μm and 5.85±0.7 (n = 49), respectively. Arbor branch-derived charcoals were rectangular or square, showing uneven edges and fiber bundles on fractured surfaces (chart 11 and 12 in Fig 2), and they had a mean L value of 169.4±45 μm, and a mean L/W of 6.09±2.1 (n = 50).Charcoals from tree leaves were elliptical or circular with uneven and relative dense edges (charts 14–20 in Fig 2), with the mean L of 131.2±14.7 μm and mean L/W of 3.06±0.6 (n = 50).

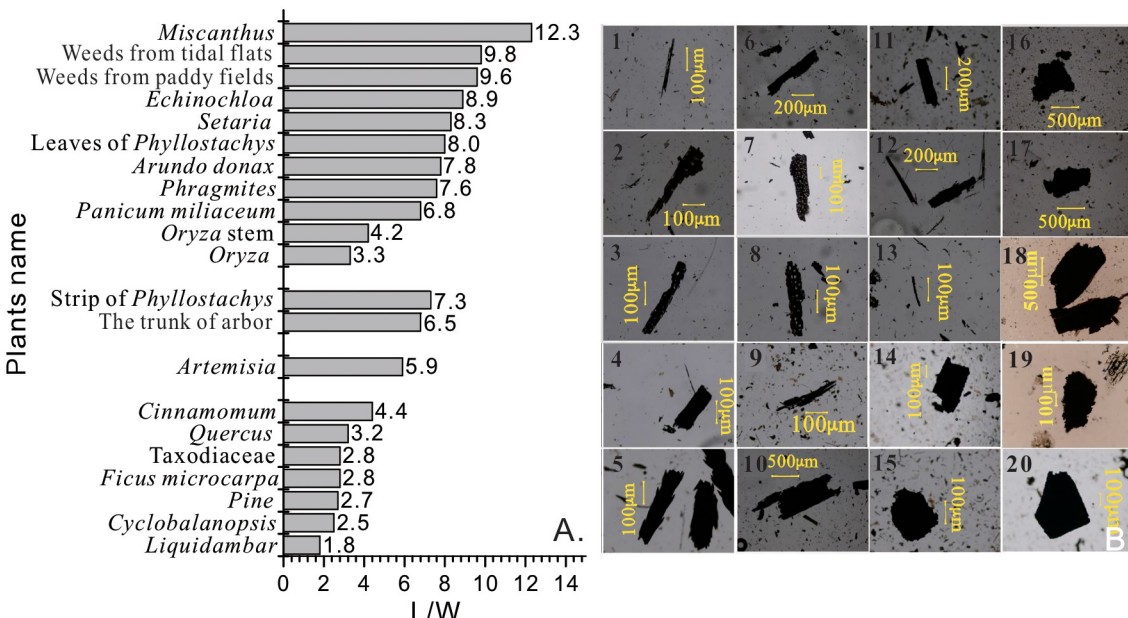

**Fig 2. L/W ratios and microscopic photographs of charcoals from combustion of modern plants in the Ningshao Plain, China.**
(A) L/W ratios (43–103 grains counted for each sample); (B) microscopic photographs of charcoal grains (1. *Miscanthus*; 2. Weeds from tidal flats; 3. Weeds from paddy fields; 4. *Echinochloa*; 5. *Setaria*; 6. Leaves of *Phyllostachys*; 7. *Arundo donax*; 8. *Phragmites*; 9. *Panicum miliaceum*; 10. *Oryza* (*Oryza* stem is not shown due to the similarity of *Oryza* stem and *Oryza*); 11. Strip of *Phyllostachys*; 12. A mixture of the trunk of arbor (including *Cinnamomum* and *Quercus*); 13. *Artemisia*; 14. *Cinnamomum*; 15. *Quercus*; 16. Taxodiaceae; 17. *Ficus microcarpa*; 18. Pine; 19.*Cyclobalanopsis*; 20. *Liquidambar*).

## Effects of fragmentation on charcoal morphology

Charcoal fragmentation gradually increased as the duration of vibration and centrifugation increased (Table 1). After vibration of 5 minutes and centrifugation of 90 minutes, the mean L of herbaceous charcoals increased from 104.1±11.5 μm (n = 41) to 108.3±5.4 μm (n = 59), and the mean L of woody charcoal decreased from 126.5±9.35 μm (n = 99) to 88.8±6.3μm (n = 59). Correspondingly, the mean L/W ratio reduced by 59.3% for herbaceous charcoals and 50.0% for woody charcoals. Although the vibration and centrifugation conditions influenced L and L/W (Table 1), the differences between herbaceous versus woody vegetation were consistent among the three conditions. Thus, we applied the method of the shortest duration (vibration for 1.5 minutes and centrifugation for 30 minutes) to all charcoals to simulate fragmentation during sedimentary processes.

The fragmentation experiments altered L and L/W of both the herbaceous and woody charcoals, but the L/W ratio changed consistently and remained larger for herbaceous charcoals than for woody charcoals (Fig 3). This observation suggests that L/W retains the original source signatures and is hence useful to distinguish the type of combusted vegetation after the sedimentation process. We calculated the mean L/W of herbaceous charcoals (4.50) and woody charcoals (1.94) and used them as the end-member values to estimate the percentages of herbaceous (%herb) *vs*. woody (%wood) vegetation in the sediment cores.

## Variations in charcoal indices in the sediment cores

The charcoal grains in the sediments show varied forms: elongated, needle-shaped, rectangular, elliptical, and circular (Fig 4). Similar forms were observed after the combustion and fragmentation experiments, suggesting charcoal grains from the sediment cores originated from

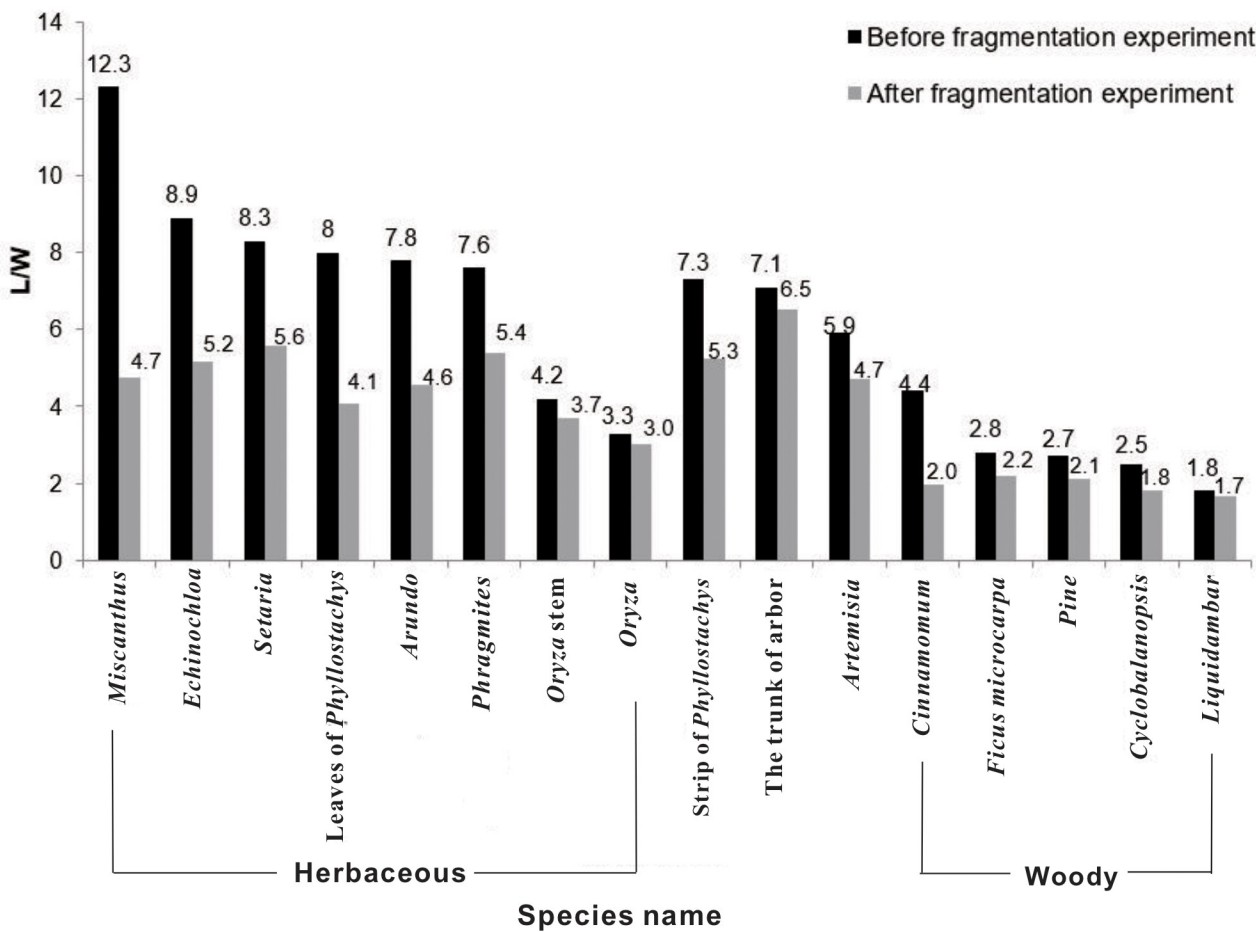

**Fig 3. Comparison of charcoal L/W ratios before and after fragmentation experiments.** Sixteen plant species from the Ningshao Plain were included in the experiments. Each bar represent data from 43–103 charcoal grains.

the combustion of a mixture of herbaceous and woody plants. Charcoal forms changed across the four sedimentary units, with typical morphological features of charcoals of different depths shown in Fig 4. Prior to the Holocene at 27.00 m, charcoals were primarily woody (arbor type), and the herbaceous count was only 7% (Fig 4A). At 18.20 m (8570 yr B.P.), round or oval woody charcoals dominated, accounting for 68% of total charcoals, and herbaceous charcoals accounted for 32% (Fig 4B). At 17.81 m (8550 yr B.P.), long, columnar herbaceous charcoals dominated, and herbaceous counts were about 88% (Fig 4C). At the depth of 4.21 m (7600 yr B.P.), long and columnar herbaceous charcoal grains were more abundant than round and oval woody charcoal grains (Fig 4D), and herbaceous charcoals accounted for about 66% of total charcoals.

Using end-member values from the fragmentation experiments, we estimated percentage contributions of combusted herbaceous versus woody vegetation upon the sedimentary sequence (Fig 5C). The results are largely consistent with those from point-counting method. According to the variation in charcoals, we identified four environmental stages (Fig 5 and Table 2). Stage I (33.00−26.00m, the last glacial period) was characterized by low charcoal concentrations (grains per gram of sediment) and low values of L and L/W, which indicates a small contribution of herbaceous charcoals varying in the range of 0−26.00%. Stage II (26.00−15.00m, 9200−8240 yr B.P.) was characterized by greater charcoal concentrations and higher

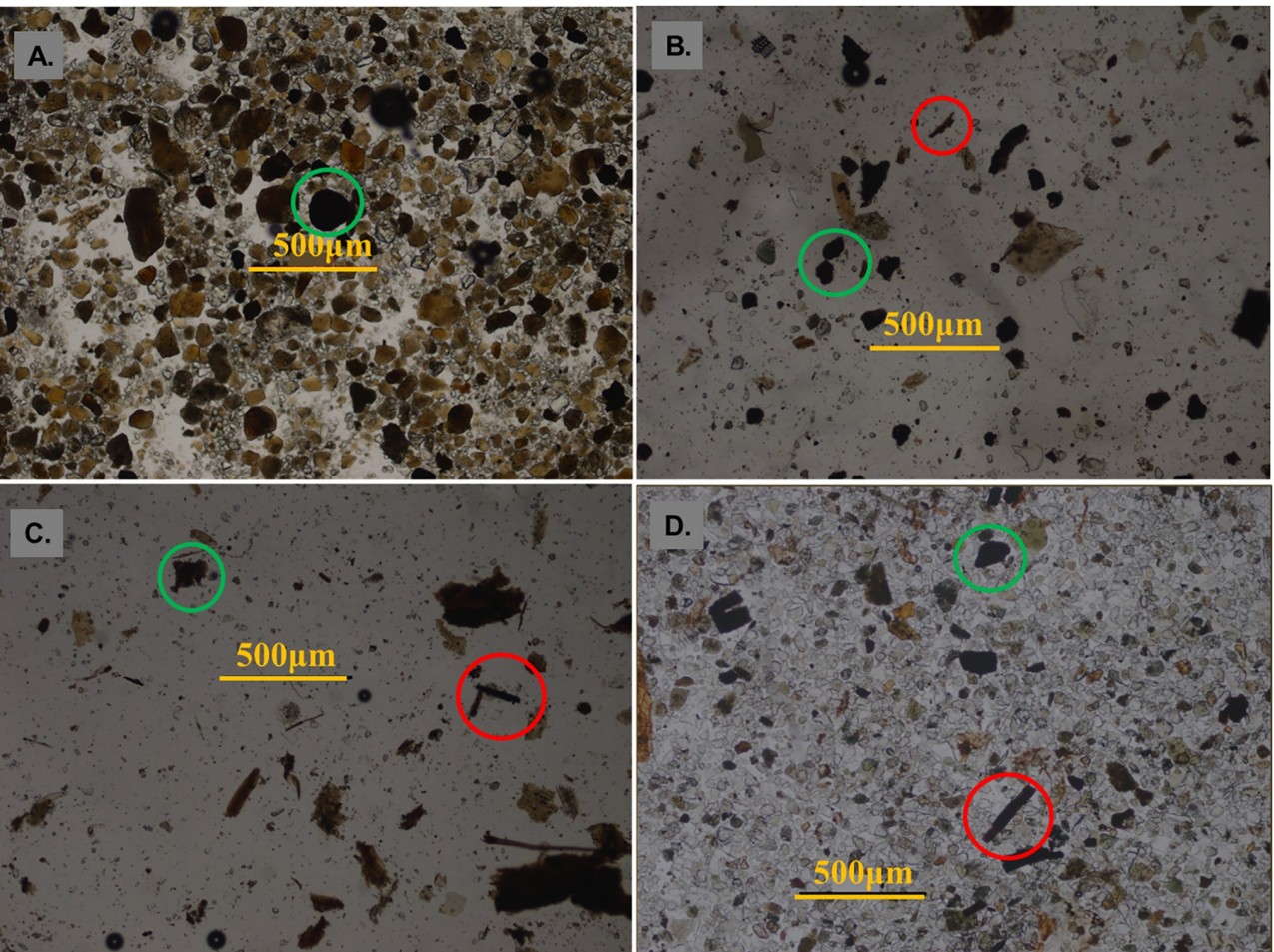

**Fig 4. Micrograph of typical charcoals from different depths of the sediment cores KHQ-14/15 from the Ningshao Plain, China.** (A) 27.00m, (B) 18.20m, (C) 17.81m and (D) 4.21m. Red circle marks herbaceous charcoal grains, and green circle marks woody charcoal grains.

values of L and L/W, and herbaceous charcoals fluctuated between 26.95−100%. Stage III (15.00−6.00m, 8240−7450 yr B.P.) included the period of the Kuahuqiao culture, and total charcoal concentrations and L and L/W of charcoal grains all decreased during this stage. Herbaceous charcoals varied from 28.90 to 65.00%, indicating reduced fire activities and a decrease in herbaceous vegetation that was combusted. During Stage IV (6.00−0m, after 7450 yr B.P.), the concentration and L of charcoals increased, yet L/W decreased further and herbaceous charcoals decreased to 1.95−46.88%. These data indicate elevated fire activities but a continuous reduction in the relative abundance of herbaceous vegetation that were combusted (Fig 5E).

## Discussion

### Charcoal morphology as an indicator of paleofire and paleovegetation

We found that the shape of charcoal grains differed systematically between herbaceous and woody plants. The anatomical structure of plants, such as the size and shape of cells and the abundance of vascular and fiber bundles, may be the primary factor determining the shape of charcoal grains [28]. Charcoals from herbaceous plants tend to break into slender fragments

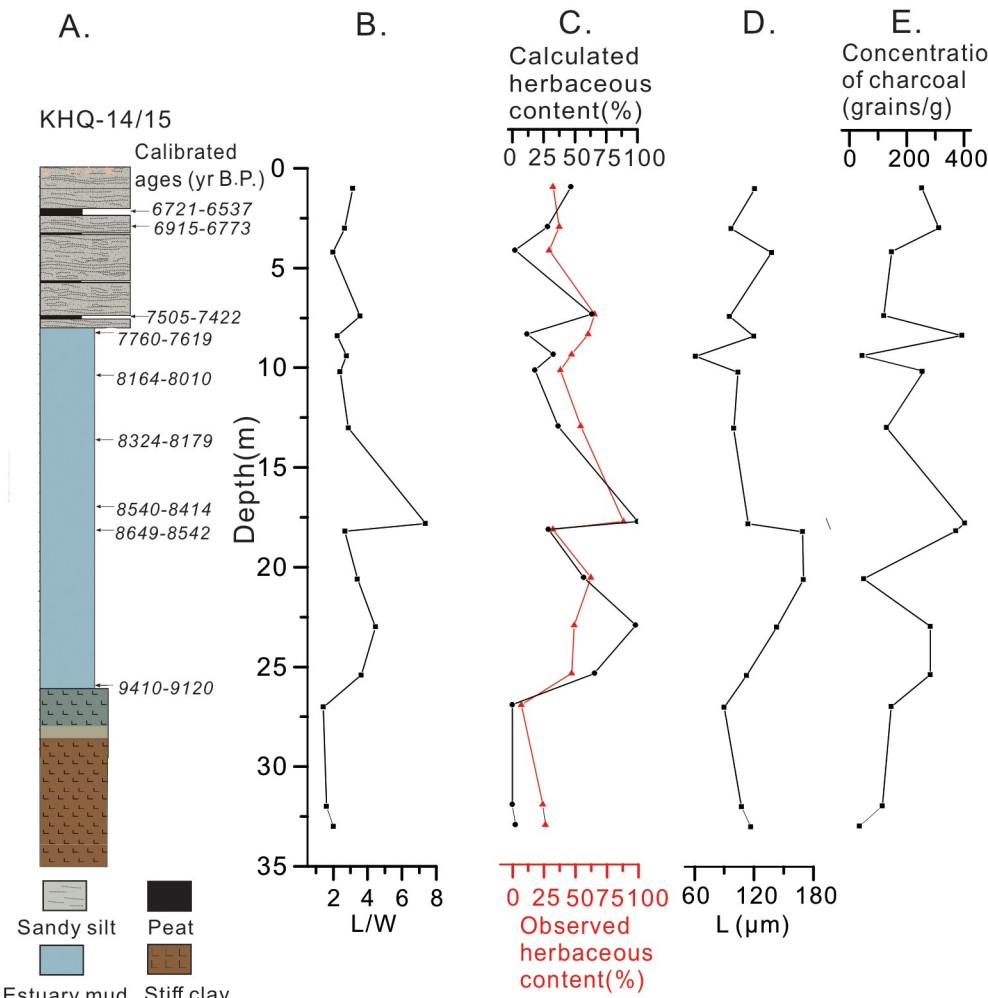

**Fig 5. Lithology and charcoal indices of the sediment cores (KHQ-14/15) from the Ningshao Plain, China.** (A) lithology; (B) mean L/W ratios of charcoal grains (each data point was calculated from 38–118 charcoal grains); (C) percent contributions of herbaceous charcoals based on end-member calculation (black line) and microscopic counting (red line); (D) mean charcoal length (each data point was calculated from 38–118 charcoal grains); (E) total charcoal concentration.

along the long axis under external forces, likely due to abundant vascular and fiber bundles in herbal plant tissues. In contrast, woody plants have a more uniform composition made of plant fibers and lignin, and their charcoals tend to break along a plane intersecting with the long axis, resulting in rectangular shapes with reduced L/W values. Leaves have

**Table 2. Temporal variation in charcoal concentration and morphological indices from the sediment cores (KHQ-14/15) from the Ningshao Plain, China.**

| Stage | Depth(m) | Calibrated ages (yr B.P.) | Mean charcoal concentration (grains/g) | Mean L(μm) | Mean L/W | Herbaceous charcoals (%) |
|-------|----------|---------------------------|----------------------------------------|------------|----------|--------------------------|
| I | 33.00−26.00 | The last glacial period | 98.70 | 104.69 | 1.68 | 0−26.00 |
| II | 26.00−14.01 | 9200−8240 | 277.66 | 141.62 | 4.31 | 28.90−100.00 |
| III | 14.01−6.00 | 8240−7450 | 187.70 | 97.50 | 2.77 | 11.72−65.00 |
| IV | 6.00−0 | After 7450 | 237.48 | 118.43 | 2.60 | 1.95−46.88 |

multidirectional veins and fibers and thus do not show a preferential breakage direction, thereby producing charcoal grains with irregular shapes and complex internal structures [28, 32].

The systematic morphological differences of charcoal grains due to vegetation types substantiate the use of end-member L/W values for reconstructing the types of paleovegetation. End-member estimation and direct microscopic counting yielded consistent results on the relative abundance of herbaceous (or woody) vegetation through the sedimentary sequence (Fig 5C). However, end-member calculation produced negative values of %herbaceous vegetation for two samples (27.00m and 31.99m), which is most likely a result of L/W variability due to species or/and depositional processes, highlighting the importance of more extensive calibration of end-member L/W values using local vegetation to generate robust results. Nevertheless, measuring charcoal L/W does not require careful examinations of morphological characteristics of individual charcoal grain and meticulous counting, thereby providing a more efficient alternative to the conventional counting method.

## Fire history in the Ningshao Plain and implications for the Kuahuqiao agricultural practices

Charcoal indices reveal changes in fire activities since the last glacial period in the Ningshao Plain. During the last glacial period (27.00–25.40m), fire activities were scarce and woody plants were the dominant type of combusted vegetation. Fire activities became more prevalent during the early Holocene (9200–8240 yr B.P.), when herbal plants were the dominant fuel for fire. During the period of 8240–7450 yr B.P. that enclosed the entire history of the Kuahuqiao culture, both fire activities and the abundance of combusted herbs decreased. After 7450 yr B. P., fire activities especially *in situ* or adjacent fires showed slight increases as evidenced by moderate increases to charcoal concentration and L values (Fig 5D), while the relative abundance of herbaceous vegetation remained relatively low. The temporal patterns in fire activities can be partially explained by the climatic and environmental context—the last glacial period was characterized by a cold climate and sparse vegetation, which are not conducive to frequent and intense wildfire, yet temperature increases during the Holocene can stimulate the growth of vegetation and associated wildfire. The dominance of herbaceous plants as the fuel source agrees well with the pollen data showing an increase in Poaceae (decrease in *Quercus* contemporaneously), which is a typical grass in the study region during the Early Holocene [5, 7].

The most distinct change throughout the sedimentary sequence is the rapid rise in the L/W of charcoals and a corresponding increase in percent herbaceous charcoals (from 27% to 100%) at 8550 yr B.P. This change marks a transition from arbor and shrub-dominated vegetation to herb-dominated vegetation. Meanwhile, the L of charcoals decreased rapidly at 8550 yr B.P., confirming the transition from woody to herbal vegetation. Phytolith analysis of the KHQ-14/15 cores [35] (Zhou et al., in preparation) also show that the content of woody phytoliths decreased yet herb phytoliths reached to the maximum (Fig 6B). In particular, the content of the *Millet subfamily* and *Bluegrass subfamily* exhibited changes consistent with those of herbaceous charcoals (Fig 6B and 6A). The pollen data of the sediment cores KHQ-14/15 also showed decreases to *Quercus* and increases to herbaceous vegetation, especially Gramineae, at 8550 yr B.P. [5] (Fig 6C). However, this observed vegetation change contradicts with the regional paleoclimatic record that shows a warm and wet climate, under which the growth of trees should be favored over that of herbs [6] (Fig 6D). For example, pollen and charcoal data of the corresponding age from lake sediments in the lower Yangtze River Basin show the flourishment of subtropical mixed forest composed of evergreen broadleaf and deciduous trees [6] and infrequent wildfire [10, 36]. The contrasting results from the present study suggest the

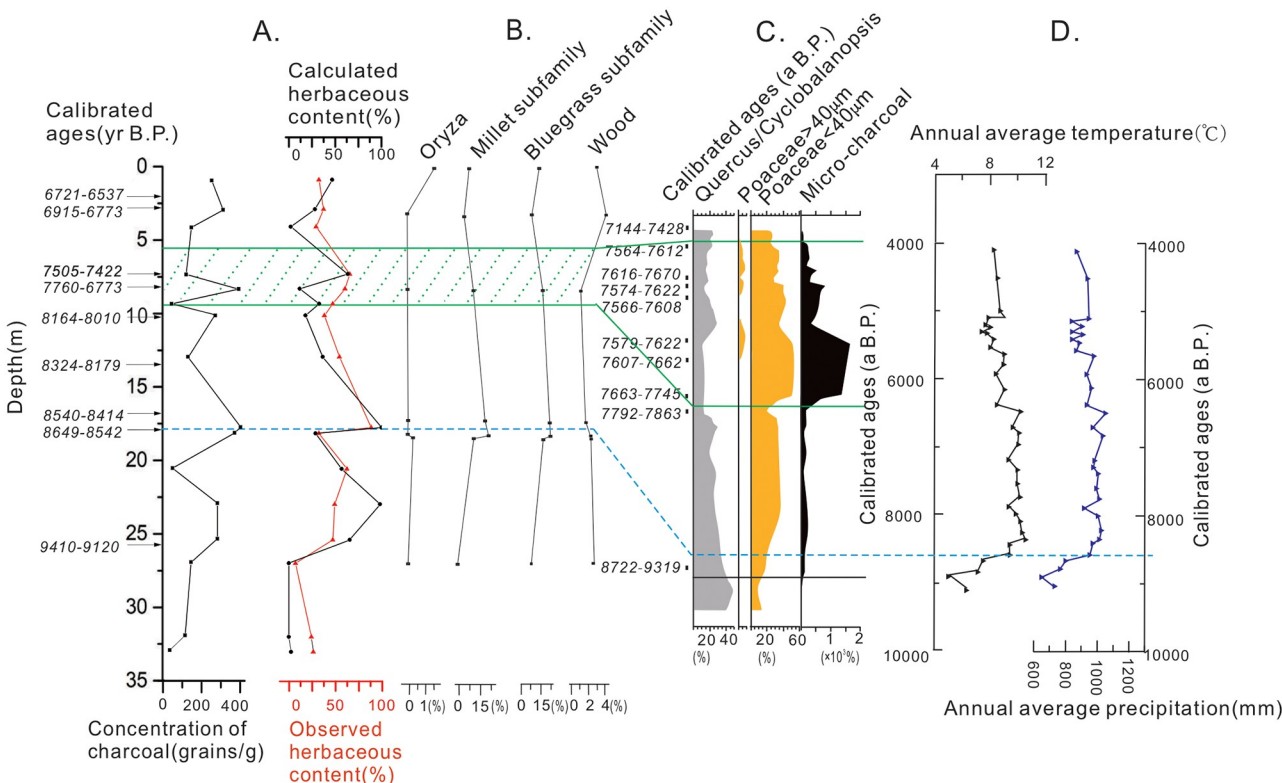

**Fig 6. Comparison of multiple proxies from sedimentary profiles around the Kuahuqiao area over the past 10k years.** (A) the charcoal indices from the sedimentary cores (KHQ-14/15) (black line indicates results from charcoal L/W end-member value calculation; red line indicates results from microscopic counting); (B) Phytolith of the sediment cores (KHQ-14/15) (under review) [35]; (C) Pollen and micro-charcoal from the Kuahuqiao archeological site [5]; (D) Reconstructed temperature and rainfall variation of the study period based on pollen records from Pingwang, Jiangsu, which is 120 km from the Kuahuqiao archeological site [6]. Green shaded areas in panel (A) and (B) outline the Kuahuqiao culture period.

significance of local vegetation and environment in mediating sedimentary charcoals. Specifically, the rise in herbaceous vegetation in our study area is most likely a result of sedimentary environmental changes to tidal flats that can support abundant growth of herbaceous plants. Similar changes have been recorded in Hemudu area during 6500–6200 yr B.P., with pollen indicating abundant aquatic herbs appeared when the sedimentary environments was supratidal flat [37]. Another observation worth noting is that rice phytoliths were found in sediments of 8550 yr B.P., along with intensive fire activities and more abundant herbaceous plants being combusted (Fig 6C). Although the common notion holds that the Kuahuqiao culture had not been developed at that time, it is possible that small-scale human activities had started earlier than previously thought [38–40].

During the Kuahuqiao culture period from 8250 yr B.P. to 7450 yr B.P., a large number of charcoal grains were found at the archaeological sites (Fig 6C), yet it remains unknown whether they originated from large-scale fire due to agricultural activities or small-scale fire due to daily life activities. Our results from sedimentary sequences that are merely ~1.5 km away from the archaeological sites show reduction in charcoal concentrations and percent herbaceous charcoals during the Kuahuqiao cultural period, indicating no constantly massive burning beyond the archaeological site. This result agrees with the charcoal record from an adjacent natural sedimentary profile [9, 41] but contrasts with the charcoal evidence from the archaeological site that indicates more intensive fire during the Kuahuqiao cultural period. This seeming discrepancy can be explained by that human-induced fires were restricted to a

small geographic area and did not generate large disturbance to surrounding areas that were not inhabited. Sedimentary records of facies and grain size from the Ningshao Plain suggests a rapid sea-level rise in the early stage of the Kuahuqiao culture, as shown by the transition to tidal flat facies between 18.40–26.30m in the sediment cores of the present study (Fig 5) and sedimentary evidence presented previously [42, 43]. Pollen data from the sediment cores of this study show that aquatic plant sporopollen such as *Typha* decreased rapidly during the cultural period [34, 41], confirming increases to both surface area and depth of water bodies. Sea-level rise can inundate agricultural fields and hinder the human effort in expanding the areas of rice cultivation. Therefore, our data do not support continuous, large-scale human fire activities beyond the archeological site, thereby providing no evidence for the use of fire to clear a large ground space to expand rice paddies during the occupation of the Kuahuqiao culture.

## Conclusion

The objective of this study was to test the hypothesis that slash-and-burn agricultural practice was adopted by the Kuahuqiao culture. We conducted laboratory combustion and fragmentation experiments on representative modern vegetation from the Ningshao Plain and established charcoal-based paleofire indices. We found systematic differences in morphology between herbaceous and woody charcoals and established the corresponding L/W end-member values, which are 4.50 for herbaceous charcoals and 1.94 for woody charcoals. These indices were then applied to wetland sediment cores adjacent to the Kuahuqiao archaeological site to reconstruct the fire and vegetation history. We found that fire activities were scarce during the last glacial period but exhibited evident increases since the Holocene. The most pronounced change occurred at about 8550 yr B.P., as marked by a sharp increase in the abundance of combusted herbaceous plants, which may reflect local environmental changes or early use of fire by humans. However, during the Kuahuqiao cultural period (8250–7450 yr B.P.), we found no evidence of large-scale fire, suggesting that the Kuahuqio ancestors did not constantly carry out large-scale slash-and-burn farming activities to expand rice cultivation to areas beyond the archeological site.

## Supporting information

**S1 Table. Details of charcoal indices from the sediment cores (KHQ-14/15).**
(XLSX)

## Acknowledgments

We thank Dr. J. Shu and another anonymous reviewer for providing constructive comments that have contributed to revisions and improvement of the manuscript. We are grateful to ChunMei Ma, ZhuJun Hu for their discussions.

## Author Contributions

**Conceptualization:** YuanFeng Hu, Bin Zhou, YueHan Lu, JianPing Zhang, SiYu Min, SiYu Xu, Qing Yang, HongBo Zheng.

**Data curation:** YuanFeng Hu, Bin Zhou, JianPing Zhang, SiYu Min, MingZhe Dai, SiYu Xu, Qing Yang, HongBo Zheng.

**Formal analysis:** YuanFeng Hu, Bin Zhou, YueHan Lu, SiYu Min, Qing Yang.

**Funding acquisition:** YuanFeng Hu, Bin Zhou, Qing Yang, HongBo Zheng.

**Investigation:** YuanFeng Hu, Bin Zhou, JianPing Zhang, SiYu Min.

**Methodology:** YuanFeng Hu, Bin Zhou, JianPing Zhang, SiYu Min, SiYu Xu.

**Project administration:** YuanFeng Hu, Bin Zhou.

**Resources:** YuanFeng Hu, Bin Zhou, Qing Yang.

**Software:** MingZhe Dai.

**Validation:** YuanFeng Hu, MingZhe Dai.

**Writing – original draft:** YuanFeng Hu, SiYu Min, MingZhe Dai.

**Writing – review & editing:** YuanFeng Hu, Bin Zhou, YueHan Lu, JianPing Zhang, Qing Yang.

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
