## [Decision Letter · Decision Letter 0]

14 Apr 2020

PONE-D-20-06598

Abundance and morphology of charcoal in sediments provide no evidence of massive slash-and-burn agriculture during the Neolithic Kuahuqiao culture, China

PLOS ONE

Dear Mrs Bin,

Thank you for submitting your manuscript to PLOS ONE. After careful consideration, we feel that it has merit but does not fully meet PLOS ONE’s publication criteria as it currently stands. Therefore, we invite you to submit a revised version of the manuscript that addresses the points raised during the review process.

all comments must be addressed before re-submission.

We would appreciate receiving your revised manuscript by May 29 2020 11:59PM. To enhance the reproducibility of your results, we recommend that if applicable you deposit your laboratory protocols in protocols.io, where a protocol can be assigned its own identifier (DOI) such that it can be cited independently in the future. For instructions see: http://journals.plos.org/plosone/s/submission-guidelines#loc-laboratory-protocols

We look forward to receiving your revised manuscript.

Kind regards,

Peter F. Biehl, PhD

Academic Editor

PLOS ONE

Additional Editor Comments (if provided):

Your manuscript has now been seen by two referees, whose comments are appended below. You will see from these comments that while the referees find your work of great interest, one reviewer has raised substantial concerns that must be addressed. In light of these comments, we cannot accept the manuscript for publication, but would be interested in considering a revised version that addresses these serious concerns.

We hope you will find the referees' comments useful as you decide how to proceed. Should presentation of further data and analysis allow you to address these criticisms, we would be happy to look at a substantially revised manuscript. However, please bear in mind that we will be reluctant to approach the referees again in the absence of major revisions.

2. In your Methods section, please provide additional information regarding the permits you obtained to collect samples for the present study. Please ensure you have included the full name of the authority that approved the field site access and, if no permits were required, a brief statement explaining why.

4. We note that Figure 1 in your submission contains map images which may be copyrighted. All PLOS content is published under the Creative Commons Attribution License (CC BY 4.0), which means that the manuscript, images, and Supporting Information files will be freely available online, and any third party is permitted to access, download, copy, distribute, and use these materials in any way, even commercially, with proper attribution. For these reasons, we cannot publish previously copyrighted maps or satellite images created using proprietary data, such as Google software (Google Maps, Street View, and Earth). For more information, see our copyright guidelines: http://journals.plos.org/plosone/s/licenses-and-copyright.

Reviewers' comments:

Reviewer's Responses to Questions

**Comments to the Author**

1. Is the manuscript technically sound, and do the data support the conclusions?

Reviewer #1: Yes

Reviewer #2: Yes

2. Has the statistical analysis been performed appropriately and rigorously? 

Reviewer #1: Yes

Reviewer #2: Yes

3. Have the authors made all data underlying the findings in their manuscript fully available?

Reviewer #1: Yes

Reviewer #2: Yes

4. Is the manuscript presented in an intelligible fashion and written in standard English?

Reviewer #1: Yes

Reviewer #2: Yes

5. Review Comments to the Author

Reviewer #1: Zhou et al.’s paper presents an interesting macrocharcoal-based evidence to argue the previous hypothesis that whether the slash-and-burn practice was used in early rice farming in the low-lying coastal Kuahuqiao area 8-7ka. The simulation experiment using common modern plants for burning successfully established the useful criteria to morphologically discriminate charcoal between herbs and woods. Their result shows that there was no regional human-induced vegetation burning for rice agriculture at Kuahuqiao, which agrees well with pollen analysis at the archaeological site. This study offers a significant insight regarding the early rice farming in coastal plains.

Several questions should be addressed before publication.

1) The incomplete selected plant list for the burning experiment

The study site is located in the monsoonal climate under which zonal broadleaved evergreen forests flourish. Based pollen results (Zong et al., 2007; Innes et al., 2009; Shu et al., 2010, 2012) show that Pinus and Cyclobalanopsis, important trees, contribute a large proportion into pollen flora. However, these two taxa are ignored in the authors’ combustion experiment. Please tell the reason why these important trees are lack.

2) Low sampling resolution

Zhou et al. utilized only 13 sediment samples for macro-charcoal analysis collected from the combined 34m-long cores. Even worse is that merely two samples from 8240-7450 cal.aBP corresponding to the cultural layers at the archaeological site. The low resolution definitely weakens the conclusion they drew to argue against the slash-and-burn hypothesis. I strongly suggest more samples are highly necessary esp. during the 8-7ka interval.

3) Abundance herbaceous charcoal around 8550 cal.aBP.

The sedimentary charcoal shows high concentration of herbaceous charcoal near the 8550 years ago. The authors provide a reasonable explanation for the exceptional high content that the tide environment encouraged herbs to colonize the tidal flat. However, is it a common situation in coastal area? Please provide published examples to support this idea in order to convince the readers.

Minor mistakes or questions:

1) lines 95-101, “northern boundary of the subtropical climate zone” could be central subtropical zone; “dominated by north subtropical evergreen deciduous ” north corrected to central; Pinus Linn, Cyclobalanopsis oerst? Correct to Pinus, Cyclobalanopsis. Gramineae to Gramineae or Line

2) Line 114, Brassica chinensis is an important economical oil-producing plant and was introduced into China during late historical time. Thus, it is impropriate to burn it for charcoal analysis.

3) Line 117, “a mixture of Wood (simplified as Wood).” Wood here is confusing. What are included? Please detail it.

4) Line 141, “seven plant residues of KHQ-1502 and two plant residues of KHQ-14 were” nine samples was dated. However, 8 ages are indicated in Fig.6, why

5）Fig.3, only 13 taxa plant are shown. However, 18 taxa are secletd for combustion experiment. Why? Moreover, Artemisia in souther China are mainly in herb form instead of trees. Also, Bamboos are usually assigned into poaceaous woods.

Reviewer #2: As a direct product of fire and vegetation, paleocharcoal is one of the most important evidence to reconstructing the fire history and ancient human activities. However, it is still controversial whether morphological features of charcoal can be used as indicators to identifying plant species. In this manuscript, the authors innovatively conducted combustion and fragmentation conditional experiments based on modern representative reference to simulate fire and depositional processes. They established morphological criteria of charcoal from modern vegetation to distinguish herbaceous and woody plants, and further applied them to sediment cores adjacent to the archaeological site revealing local environment changes and early human activities, providing new insight about slash-and-burn farming activities in lower Yangtze region in the early Holocene. This case study provides more evidence to further illustrating the relationship between the production, deposition of charcoal and farming activity in the early stage of agriculture.

The word in line 29 "then applied to sediment cores (KQH-14/15)", "KQH" should be "KHQ"?

6. PLOS authors have the option to publish the peer review history of their article (what does this mean?). If published, this will include your full peer review and any attached files.

Reviewer #1: Yes: Junwu SHU

Reviewer #2: No

---

## [Author Response · Author response to Decision Letter 0]

11 Jul 2020

Dear Dr. Biehl,

We appreciate the time that you took in addressing our manuscript. In our revision we have addressed all the main concerns of the reviewers and have made the requested specific changes based on the reviewers’ comments. Specifically, we have performed more experiments to collect additional data on plant taxa and time period of interest. Furthermore, we have made extensive edits to improve the readability of the manuscript. You will see our responses to the comments below. We appreciate these suggestions and think they significantly improve the manuscript and hope that you find that our manuscript is now appropriate for publication. 

We thank you for your time and help in making this paper much clearer to readers and look forward to hearing from you. We are happy to make additional revisions if necessary.

Sincerely,

Bin Zhou

Review Comments to the Author

Reviewer #1: Zhou et al.’s paper presents an interesting macrocharcoal-based evidence to argue the previous hypothesis that whether the slash-and-burn practice was used in early rice farming in the low-lying coastal Kuahuqiao area 8-7ka. The simulation experiment using common modern plants for burning successfully established the useful criteria to morphologically discriminate charcoal between herbs and woods. Their result shows that there was no regional human-induced vegetation burning for rice agriculture at Kuahuqiao, which agrees well with pollen analysis at the archaeological site. This study offers a significant insight regarding the early rice farming in coastal plains.

Reply: We thank the reviewer for taking the time to evaluate our manuscript and providing constructive comments, which contributed to the improvement of the manuscript. 

Several questions should be addressed before publication.

1) The incomplete selected plant list for the burning experiment

The study site is located in the monsoonal climate under which zonal broadleaved evergreen forests flourish. Based pollen results (Zong et al., 2007; Innes et al., 2009; Shu et al., 2010, 2012) show that Pinus and Cyclobalanopsis, important trees, contribute a large proportion into pollen flora. However, these two taxa are ignored in the authors’ combustion experiment. Please tell the reason why these important trees are lack.

Reply: We thank the reviewer for this suggestion. We have performed additional combustion experiments and included Pine, Cyclobalanopsis, Oryza stem. 

2) Low sampling resolution

Zhou et al. utilized only 13 sediment samples for macro-charcoal analysis collected from the combined 34m-long cores. Even worse is that merely two samples from 8240-7450 cal.aBP corresponding to the cultural layers at the archaeological site. The low resolution definitely weakens the conclusion they drew to argue against the slash-and-burn hypothesis. I strongly suggest more samples are highly necessary esp. during the 8-7ka interval.

Reply: We have added three additional data points during the 8-7 ka, and the result is now shown in the revised manuscript. These new results did not change our conclusion. 

3) Abundance herbaceous charcoal around 8550 cal. a BP.

The sedimentary charcoal shows high concentration of herbaceous charcoal near the 8550 years ago. The authors provide a reasonable explanation for the exceptional high content that the tide environment encouraged herbs to colonize the tidal flat. However, is it a common situation in coastal area? Please provide published examples to support this idea in order to convince the readers.

Reply: We thank the reviewer for this suggestion and have now added references to support this idea. Please check lines of 367-369, and reference 37. 

Minor mistakes or questions:

1) lines 95-101, “northern boundary of the subtropical climate zone” could be central subtropical zone; “dominated by north subtropical evergreen deciduous ” north corrected to central; Pinus Linn, Cyclobalanopsis oerst? Correct to Pinus, Cyclobalanopsis. Gramineae to Gramineae or Line

Reply: We have made the correction in lines 95-101.

2) Line 114, Brassica chinensis is an important economical oil-producing plant and was introduced into China during late historical time. Thus, it is impropriate to burn it for charcoal analysis.

Reply: We meant “Weeds from tidal flats”. We apologize for this error and have made the modification in lines 114. 

3) Line 117, “a mixture of Wood (simplified as Wood).” Wood here is confusing. What are included? Please detail it.

Reply: We have added the details— “the trunk of arbor including Cinnamomum and Quercus”. Modification has been made in the figures and article.

4) Line 141, “seven plant residues of KHQ-1502 and two plant residues of KHQ-14 were” nine samples was dated. However, 8 ages are indicated in Fig.6, why

Reply: We apologize for the omission. We have added the age data to Fig.6.

5）Fig.3, only 13 taxa plant are shown. However, 18 taxa are selectedd for combustion experiment. Why? Moreover, Artemisia in southern China are mainly in herb form instead of trees. Also, Bamboos are usually assigned into poaceaous woods.

Reply: We have added three additional taxa for the combustion and fragmentation experiments. We performed fragmentation experiments on the majority (16 out of 21) but not all of selected taxa because fragmentation showed relatively consistent effects on all samples. We included the major and representative taxa of the study area, which should be sufficient to address the research objective.

Following this suggestion, we assigned Phyllostachys as the trunk of woods. Because the Artemisia we collected in the field were tall and their stems have become lignified, we classified it as a shrub according to botanical definition.

Reviewer #2: As a direct product of fire and vegetation, paleocharcoal is one of the most important evidence to reconstructing the fire history and ancient human activities. However, it is still controversial whether morphological features of charcoal can be used as indicators to identifying plant species. In this manuscript, the authors innovatively conducted combustion and fragmentation conditional experiments based on modern representative reference to simulate fire and depositional processes. They established morphological criteria of charcoal from modern vegetation to distinguish herbaceous and woody plants, and further applied them to sediment cores adjacent to the archaeological site revealing local environment changes and early human activities, providing new insight about slash-and-burn farming activities in lower Yangtze region in the early Holocene. This case study provides more evidence to further illustrating the relationship between the production, deposition of charcoal and farming activity in the early stage of agriculture.

The word in line 29 "then applied to sediment cores (KQH-14/15)", "KQH" should be "KHQ"?

Reply: We thank the reviewer for taking the time to evaluate our manuscript and providing constructive comments, which have helped to improve the manuscript. We have made the correction in line 29.

---

## [Decision Letter · Decision Letter 1]

30 Jul 2020

Abundance and morphology of charcoal in sediments provide no evidence of massive slash-and-burn agriculture during the Neolithic Kuahuqiao culture, China

PONE-D-20-06598R1

Dear Dr. Bin,

We’re pleased to inform you that your manuscript has been judged scientifically suitable for publication and will be formally accepted for publication once it meets all outstanding technical requirements.

Kind regards,

Peter F. Biehl, PhD

Academic Editor

PLOS ONE

Additional Editor Comments (optional):

Reviewers' comments:

Reviewer's Responses to Questions

**Comments to the Author**

1. If the authors have adequately addressed your comments raised in a previous round of review and you feel that this manuscript is now acceptable for publication, you may indicate that here to bypass the “Comments to the Author” section, enter your conflict of interest statement in the “Confidential to Editor” section, and submit your "Accept" recommendation.

Reviewer #1: All comments have been addressed

2. Is the manuscript technically sound, and do the data support the conclusions?

Reviewer #1: Yes

3. Has the statistical analysis been performed appropriately and rigorously? 

Reviewer #1: Yes

4. Have the authors made all data underlying the findings in their manuscript fully available?

Reviewer #1: Yes

5. Is the manuscript presented in an intelligible fashion and written in standard English?

Reviewer #1: Yes

6. Review Comments to the Author

Reviewer #1: Minor corrections:

Family names should be NOT in italics in texts, such as in L99 Gramineae； L133，L162， L175，L215 and L232 Taxodiaceae；L344 Poaceae.

7. PLOS authors have the option to publish the peer review history of their article (what does this mean?). If published, this will include your full peer review and any attached files.

Reviewer #1: No

---

## [Editor Report · Acceptance letter]

7 Aug 2020

PONE-D-20-06598R1 

Abundance and morphology of charcoal in sediments provide no evidence of massive slash-and-burn agriculture during the Neolithic Kuahuqiao culture, China 

Dear Dr. Zhou:

I'm pleased to inform you that your manuscript has been deemed suitable for publication in PLOS ONE. Congratulations! Your manuscript is now with our production department. 

Kind regards, 

on behalf of

Dr. Peter F. Biehl 

Academic Editor

PLOS ONE